# Strain-Specific Effects of Biochar and Its Water-Soluble Compounds on Bacterial Growth

**Fan Yang [1], Yue Zhou [1,†], Weiming Liu [1,†], Wenzhu Tang [1], Jun Meng [2], Wenfu Chen [2] and Xianzhen Li [1,2,*]**

1   School of Biological Engineering, Dalian Polytechnic University, Dalian 116034, China
2   BETR Centre, Shenyang Agricultural University, Shenyang 110866, China
*   Correspondence: xianzhen@mail.com; Tel.: +86-(411)-8632-3717
†   These two authors contributed equally to this work.

**Abstract:** Previous studies have revealed that biochar could induce the disturbance of a microbial community above the family level. So far, very little is known about how individual bacteria are affected by biochar at genus or species levels. In this study, effects of biochar and its water-soluble compounds on the growth of individual soil bacteria were examined. Biochar derived from different feedstock showed disproportionate impacts on bacterial growth. Corncob biochar could significantly stimulate the growth of most tested strains, whereas the growth of four strains, including *Bacillus pumilus* ACCC04306 (Agricultural Culture Collection of China, ACCC), *B. licheniformis*, *B. cereus*, and *Kitasatospora viridis*, were inhibited by addition of rice husk biochar. All the biochars greatly supported the growth of *B. mucilaginosus* but inhibited that of *K. viridis*. More importantly, different strains exhibited discrepant growth response towards the same biochar sample, even when strains belong to the same species, suggesting that the effect of biochar on bacteria growth is strain-specific. Corncob biochar showed the strongest adsorption on *B. thuringiensis* but the greatest growth promotion was observed in *B. mucilaginosus*, indicating that the porous structure of biochar is not the sole factor that influences cell growth. Due to the possible stimulation or inhibition of water-soluble compounds existing in biochar, the growth variation of tested strains decreased or increased correspondingly when the washed biochar was applied, indicating that water-soluble compounds in fresh biochar play an important role in cell growth and such effect is also strain-dependent. Biochar application could also enhance potassium-/phosphate-solubilizing activities through promoting bacterial growth. All these results suggested that biochar might influence bacterial growth under different mechanisms. Our findings should be valuable for an in-depth understanding of the potential mechanism of soil bacteria changes following biochar incorporation and for biochar application in agriculture.

**Keywords:** biochar; water-soluble compounds; bacterial growth; strain-specific

## 1. Introduction

Biochar is a kind of solid carbonaceous residue produced from low-temperature pyrolysis of biomass under oxygen-free or low oxygen conditions. Biochar has distinct physico-chemical properties (i.e., highly condensed aromatic structures, hydraulic conductivity and cation exchange capacity), which can vary depending on the feedstock [1]. Recently, biochar soil amendment has received a substantial amount of attention because biochar could increase crop yield and enhance soil carbon sequestration [2]. Several factors have been proposed to impact crop production, such as soluble nutrients contained in the fresh biochar, the sorption of growth-inhibiting substances by biochar, the retention of dissolved organic matter in soil, and the ameliorated soil properties [3]. However,

a growing body of evidence suggests that the enhancement of crop production is correlated with biochar-induced changes in soil biota, a component of the terrestrial ecosystem that plays a key role in nutrient cycling [4].

Biochar-induced stimulation of microbial activities, along with changes in microbial community composition and structure, has also been reported [5]. It seems that fresh biochar supplies labile organic carbon fractions for soil biota, whereas the direct evidence for the transfer of available carbon and nutrients from biochar to microorganisms is still lacking [6]. Moreover, chemical compounds that inhibit microbial activities have also been found in biochar [7]. Other studies have indicated that biochar supplies a habitat for soil microbes [8], but so far only a few microscopic studies have directly observed microbial colonization in biochar and the reason for this process remained uncertain [6]. Actually, it is unlikely that these biochar-induced changes spread equally across different phylotypes or functional groups [9].

Biochar has been reported to positively or negatively affect soil microbial abundance [10]. The Illumina sequencing of soil bacterial 16S rRNA genes suggested that the relative abundance of Acidobacteria (phylum), Chloroflexi (phylum), and Gemmatimonadetes (phylum) decreased but that of Proteobacteria (phylum), Bacteroidetes (phylum), and Actinobacteria (phylum) increased after biochar soil amendment [11]. A pot experiment showed that biochar had a positive influence on the abundance of Bradyrhizobiaceae (family), Hyphomicrobiaceae (family), Streptosporangineae (family), and Thermomonosporaceae (family), but had a negative effect on the abundance of Streptomycetaceae (family) and Micromonosporaceae (family) [12]. So far, almost all the studies have focused on the changes in microbial community above the family level, while there has been little information about the effect of biochar on bacteria at genus or species levels. Furthermore, very little is known about how individual bacteria are affected by biochar. In this study, influences of biochar and its water-soluble compounds on the growth of individual bacteria were examined in the artificial culture, and the results suggested that biochar together with its water-soluble compounds had a strain-specific impact on bacteria growth.

## 2. Materials and Methods

### 2.1. Materials, Strains and Culture Condition

Potash feldspar powder was purchased from Dabieshan Mining Co., Wuhan, China, which was sieved at a 0.149 mm size and washed with distilled water.

*Ochrobactrum* sp. ACCC10085, *Bacillus pumilus* ACCC04306, *Bacillus pumilus* ACCC01736, *Bacillus licheniformis* ACCC01050, *Azotobacter chroococcum* ACCC01077, *Pseudomonas* sp. ACCC02568, *Bacillus megaterium* ACCC01667, *Bacillus cereus* ACCC04289, *Kitasatospora viridis* ACCC02567, and *Bacillus thuringiensis* ACCC04311 were purchased from the Agricultural Culture Collection of China (ACCC) (Beijing China). *Bacillus mucilaginosus* AS1153 and *B. megaterium* AS1217 were purchased from the China General Microbiological Culture Collection Centre (Beijing, China). All tested strains were incubated at 30 °C and 150 rpm in the nutrient broth, except *Azotobacter chroococcum* that was incubated in the *Azotobacter* medium under the same culture condition.

The nutrient broth was composed of (per liter) 5 g peptone, 3 g beef extract, and 5 g NaCl at pH 7.0. The *Azotobacter* medium contained (per liter) 10 g mannitol, 0.5 g $K_2HPO_4 \cdot 3H_2O$, 0.2 g NaCl, 1 g $CaCO_3$, 0.2 g $MgSO_4 \cdot 7H_2O$ at pH 7.0–7.2. All the reagents were purchased from Shenggong Co., Shanghai, China.

### 2.2. Preparation and Physico-Chemical Analysis of Biochar

Biochar was produced from corncob, rice husk, and bamboo (purchased from local markets, Dalian, China), respectively, with pyrolysis method as described elsewhere [13], and lightly crushed and sieved to obtain a uniform particle size of 0.25–2.0 mm. The total organic carbon content of biochar was determined by total organic carbon analyzer (TOC-L, Shimadzu, Kyoto, Japan). The total

nitrogen of biochar was analyzed with the Kjeldahl method [14]. The pH of biochar was measured in water slurry at a 1:10 (*w/v*) ratio. The ash content of biochar was estimated using thermogravimetric analyzer (TGA/SDTA851, Mettler-Toledo, Schwarzenbach, Switzerland). The temperature increased at a rate of 10 °C/min (25 to 1000 °C) under air atmosphere with a flow rate of 50 mL/min [15]. Brunauer–Emmett–Teller (BET) surface area of biochar with or without bacterial adsorption was determined by $N_2$ sorption analysis at 77 K in a surface analyzer (ASAP2020, Micromeritic Instrument Corporation, Norcross, GA, USA) after degassing. Scanning electron microscopy (SEM) analysis of biochar with or without bacterial adsorption was performed using a TM3000 scanning electron microscope (HITACHI, Tokyo, Japan).

## 2.3. Preparation of Water-Soluble Compounds from Biochar

A total of 20 g of biochar was mixed with 200 mL distilled water and stirred at room temperature for 6 h. The biochar washing was repeated for four times; the washed biochar was dried at 105 °C overnight for use. The effluent from biochar was collected together and freeze-dried for subsequent experiments.

## 2.4. Effect of Biochar on Bacterial Growth

After incubation in the nutrient broth or the *Azotobacter* medium at 30 °C and 150 rpm overnight, the tested bacteria were respectively inoculated at 1% inoculum in the same medium supplemented with 0.6% washed- or unwashed-biochar. After incubated at 30 °C and 150 rpm for 12 h, the cell count (*C*) in total was determined by the colorimetric 3-(4,5-Dimethyl-2-thiazolyl)-2,5-diphenyltetrazolium Bromide (MTT) method [16]. The control was carried out in the nutrient broth or the *Azotobacter* medium without biochar. The variation in bacterial growth affected by biochar was calculated as:

$$\text{Variation (\%)} = 100 \times (C_{bio} - C_{ck})/C_{ck}, \tag{1}$$

in which the cell count in the culture with biochar addition was expressed as $C_{bio}$, the cell count in the culture without biochar was expressed as $C_{ck}$.

## 2.5. Adsorption of Bacterial Cells to Biochar

Overnight cultures of the tested strains (10 mL) were mixed with the sterilized biochar (60 mg), respectively, and incubated at 30 °C and 150 rpm for 3 h. After the biochar was collected and washed with distilled water, the bacterial count adsorbed in biochar was determined by the colorimetric MTT method [16]. The adsorption of bacterial cells in biochar was calculated as:

$$\text{Adsorption (\%)} = 100 \times C_{ads}/C_t, \tag{2}$$

in which $C_{ads}$ represents the cell count adsorbed in biochar, $C_t$ represents the total cell count in the culture fluid.

## 2.6. Effect of Biochar Adsorption on Bacterial Growth Activity

The overnight culture of *B. mucilaginosus* was inoculated with 1% inoculum in the nutrient broth containing 0.6% washed corncob biochar and incubated at 30 °C and 150 rpm for 9 h. After being washed with sterilized saline, the biochar-adsorbed cells were inoculated in the fresh nutrient broth and incubated at 30 °C and 150 rpm. The bacterial count was determined at an interval time by the colorimetric MTT method [16]. The control was carried out by incubating *B. mucilaginosus* in the nutrient broth with the overnight-cultured free cells as an inoculum.

## 2.7. Influence of Water-Soluble Compounds on Bacterial Growth

The overnight cultures of the tested strains were respectively inoculated with 1% inoculum in the nutrient broth or the *Azotobacter* medium supplemented with 0.6% of water-soluble compounds.

After incubation at 30 °C and 150 rpm for 12 h, the cell count was measured by the colorimetric MTT method [16]. The control was performed in the nutrient broth or the *Azotobacter* medium without water-soluble compounds. The variation in bacterial growth affected by water-soluble compounds was calculated as:

$$\text{Variation (\%)} = 100 \times (C_{ash} - C_{ck})/C_{ck}, \tag{3}$$

in which the cell count in the culture with water-soluble compounds was expressed as $C_{ash}$, the cell count in the culture without water-soluble compounds was expressed as $C_{ck}$.

### 2.8. Effect of Biochar on Potassium/Phosphate Solubilizing Strains

The overnight culture of *B. mucilaginosus* was inoculated at 1% inoculum in the potassium-solubilizing medium with or without 0.6% biochar derived from corncob and incubated at 30 °C and 150 rpm for 5 days. Potassium concentration in the culture supernatant was determined using an atomic absorption spectrometer (Z-8100, Hitachi, Tokyo, Japan) [17]. Potassium-solubilizing medium contained (per liter) 5 g sucrose, 2 g $Na_2HPO_4$, 0.5 g $MgSO_4 \cdot 7H_2O$, 0.005 g $FeCl_3$, 0.1 g $CaCO_3$, and 10 g potash feldspar powder at pH 7.0–7.5.

Phosphate solubilizing medium with or without 0.6% biochar supplementation was inoculated respectively with 1% inoculum of *Ochrobactrum* sp. and *B. megaterium* AS1217. Phosphate content was determined with molybdenum blue method [18] after fermentation at 30 °C and 150 rpm for 5 days. Phosphate-solubilizing medium consisted of (per liter) 5 g $MgCl_2 \cdot 6H_2O$, 0.25 g $MgSO_4 \cdot 7H_2O$, 0.2 g KCl, 0.1 g $(NH_4)_2SO_4$, 5 g $Ca_3(PO_4)_2$, and 10 g glucose. All the reagents were purchased from Shenggong Co., Shanghai, China.

### 2.9. Statistic Analysis

All experiments and assays were performed in triplicate unless otherwise stated and the data was reported as the mean ± standard deviation.

## 3. Results

### 3.1. Impact of Different Biochars on Bacterial Growth

It is known that biochars could induce the changes in bacterial community structure, but how biochars influence the cell growth of individual strain was still unclear [1]. Therefore, 12 rhizosphere-associated bacterial strains from 4 phyla were selected to examine the impact of biochar on the cell growth, these strains could subsequently enhance soil fertility through phosphate/potassium solubilization, azotification, and production of cellulase, xylanase, β-glucanase, phytase, and chitinase. Three biochar samples, including rice husk biochar, corncob biochar, and bamboo biochar, were applied in this work, and the physico-chemical properties of these biochars were listed in Table 1. All the properties varied greatly among the three biochar samples, indicating different influences that these biochars might exert on bacteria. It is worth mentioning that the average pore size of bamboo biochar was too large to be detected by BET analyzer. As shown in Figure 1, the maximal growth promotion and inhibition were observed on *B. mucilaginosus* and *K. viridis,* respectively, no matter which biochar sample was applied. However, different growth variation was detected in the bacteria culture added with biochars derived from different feedstocks. Compared with the other two biochar samples, corncob biochar showed a positive impact on all tested bacteria except *K. viridis*, and strains with the growth promotion degree more than 20% included *Ochrobactrum* sp., *B. pumilus*, *B. licheniformis*, *B. mucilaginosus*, and *B. megaterium*. Bamboo biochar had a similar behavior as corncob biochar, except its negative effect on *B. cereus*. When rice husk biochar was added, the growth of *B. pumilus* (ACCC04306), *B. licheniformis*, *B. cereus*, and *K. viridis* was suppressed, respectively. Meanwhile, different strains from the same species, such as *B. pumilus* ACCC04063 vs. ACCC011736 and *B. megaterium* AS1217 vs. ACCC01667, exhibited discrepant responses towards the same biochar. Considering the positive impact on most of tested bacteria, the biochar derived from corncob was used in the following studies.

**Table 1.** The physico-chemical properties of biochars from different raw materials.

| Property | Rice Husk Biochar | Corncob Biochar | Bamboo Biochar |
|---|---|---|---|
| pH | 10.01 | 9.64 | 9.27 |
| TN (g·kg$^{-1}$) | 6.68 | 12.69 | 9.17 |
| TOC (g·kg$^{-1}$) | 460.50 | 541.00 | 859.80 |
| Volatile matter (%) | 16.45 | 21.94 | 13.90 |
| Ash content (%) | 42.75 | 18.85 | 0.66 |
| Surface area (m$^2$·g$^{-1}$) | 92.52 | 58.38 | 0.11 |
| Average pore size (nm) | 2.72 | 3.34 | — |

"—": Not detected; TN: total nitrogen; TOC: total organic carbon.

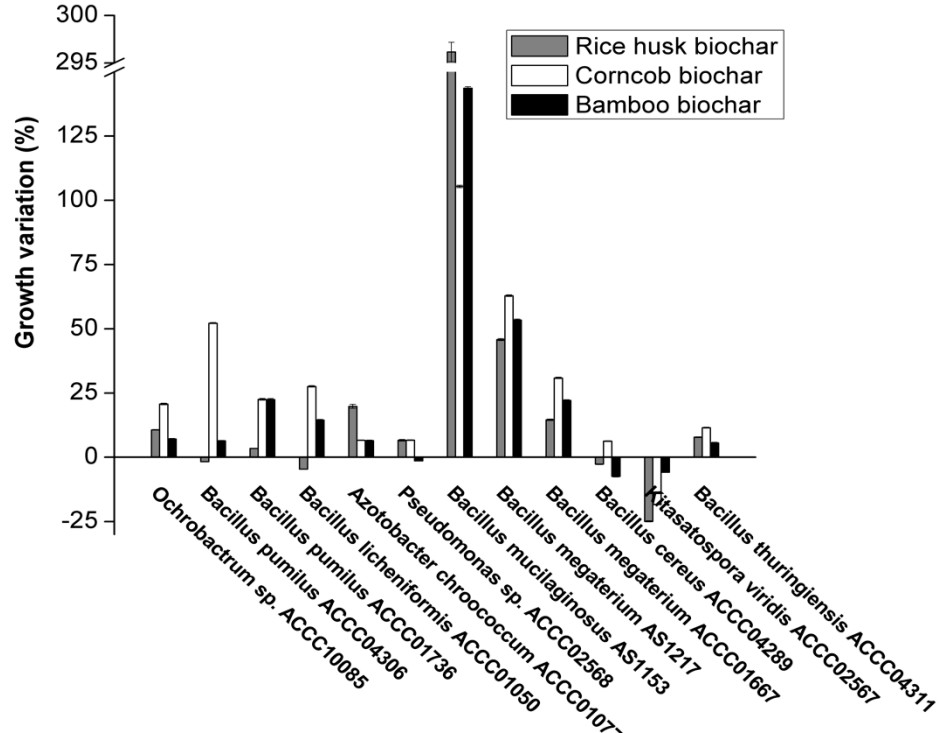

**Figure 1.** Impact of biochars derived from different feedstock on bacterial growth in the artificial culturing medium.

### 3.2. The Correlation between Adsorption Ability of Biochar and Bacterial Growth

Considering that the maximal promotion on cell growth by biochar was observed on *B. mucilaginosus* (Figure 1), the growth activity of *B. mucilaginosus* was examined after its inoculum was treated by corncob biochar. To confirm the cell adsorption, the BET surface area and SEM image of corncob biochar with or without bacteria adsorption were determined. As can be seen from the SEM image in Figure 2, the surface of untreated corncob biochar was entirely different from that of cell-adsorbed biochar. Many holes were regularly distributed in untreated biochar; however, after incubation with *B. mucilaginosus* culture, the aggregation of many particles on the surface of biochar made it rugged. BET surface area of cell-adsorbed biochar dramatically decreased to 7.9153 m$^2$·g$^{-1}$. All the results indicated that *B. mucilaginosus* cells existed on the surface of corncob biochar. After *B. mucilaginosus* was adsorbed with biochar during the seed culture process, biochar-adsorbed cells were incubated in fresh nutrient broth without biochar and the cell growth curve was detected. As shown in Figure 3, significantly different growth profiles were observed in *B. mucilaginosus* fermentation cultures incubated with biochar-treated seed and untreated seed. Compared to the culture incubated with untreated seed, *B. mucilaginosus* grew more rapidly with the biochar-adsorbed cells as the inoculum,

and the biomass was increased by 48.3% when *B. mucilaginosus* was fermented for 18 h. The result indicated that cells released from biochar gained better growth activity than free cells did. The similar promotion in growth activity was also observed differentially on the other tested bacteria (data not shown) [19].

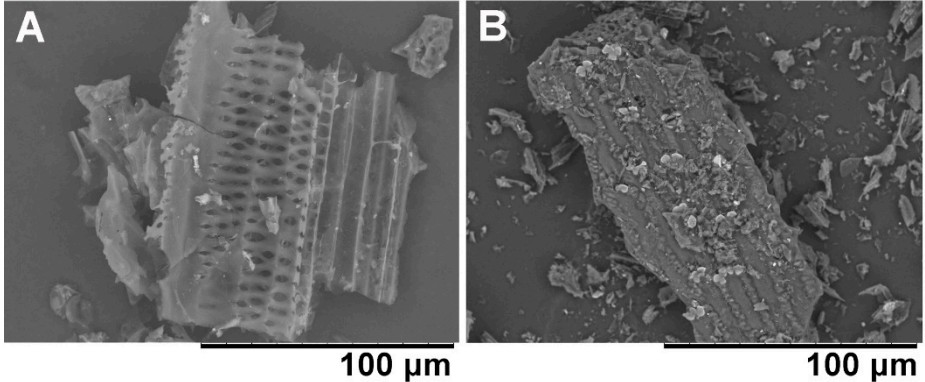

**Figure 2.** Scanning electron microscope (SEM) images of corncob biochar. (**A**) SEM image of untreated biochar (control); (**B**) SEM image of biochar adsorbed with *Bacillus mucilaginosus* cells.

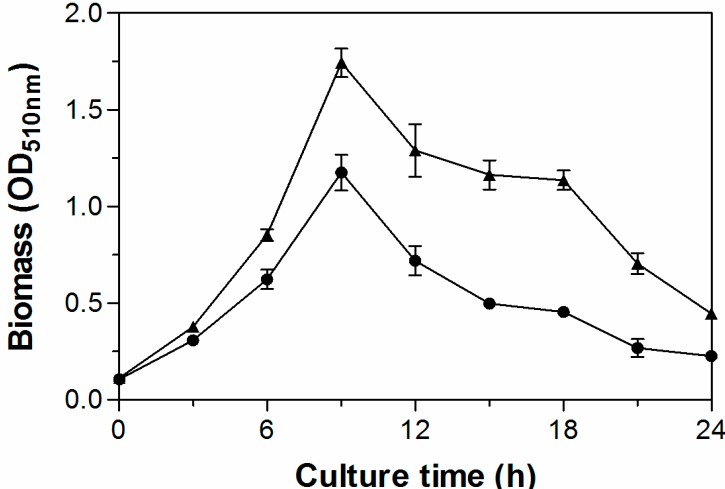

**Figure 3.** The growth curve of *Bacillus mucilaginosus* in the nutrient broth with biochar-adsorbed cells (▲) or free cells (●) as inoculum.

It has been suggested that the large surface area and porous structure of biochar could supply a favorable habitat for microbial colonization [6,20]. Thus, the correlation between the bacterial-adsorption ability of biochar and the promotion of cell growth was examined. As shown in Figure 4, the corncob biochar showed the strongest adsorption on *B. thuringiensis*, and great adsorption on *B. cereus*, *B. megaterium* (AS1217), *B. mucilaginosus*, and *Ochrobactrum* sp., but little adsorption on *A. chroococcum*. However, the variation trend in bacterial growth was not in accordance with that observed in bacterial adsorption. As for the strain *K. viridis*, the strong adsorption by biochar could also be observed, however, its growth was dramatically suppressed by the addition of biochar. Thus, the adsorption of corncob biochar towards bacteria probably is not the sole factor affecting bacterial growth. Other factors, such as chemical compounds and nutrients in biochar and water retention potential of biochar, etc., might also influence bacterial growth.

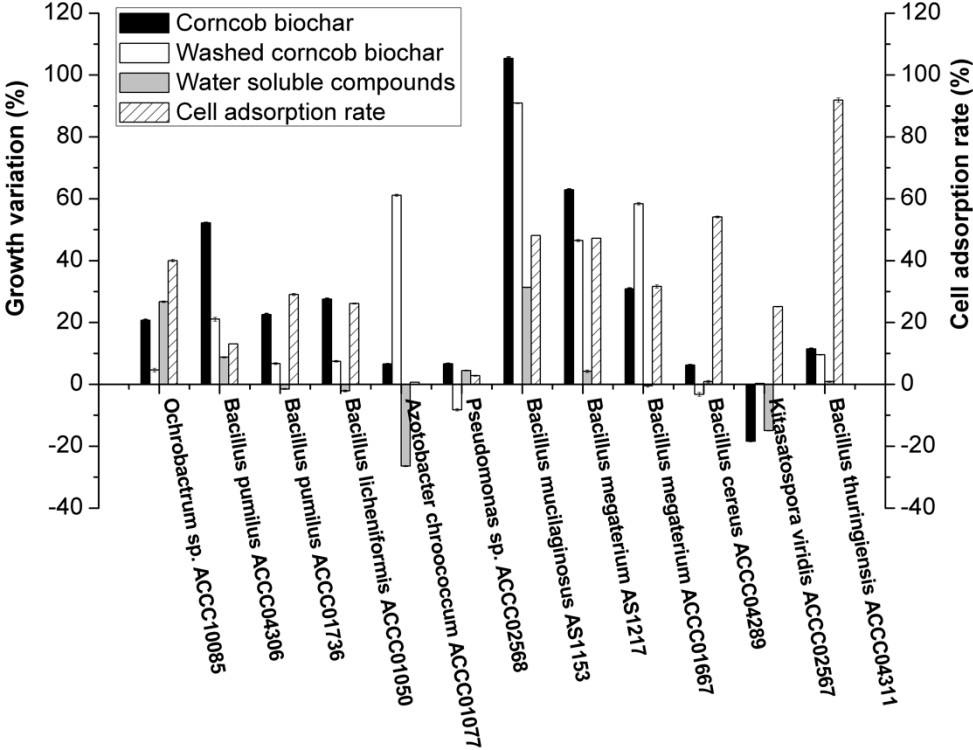

**Figure 4.** Impact of corncob biochar adsorption and its water-soluble compounds on bacterial growth in the artificial medium.

### 3.3. Effect of Water-Soluble Compounds in Biochar on Bacterial Growth

The impact of water-soluble compounds in biochar on bacterial growth was then investigated. As shown in Figure 4, the water-soluble compounds washed from biochar showed a positive or negative influence on bacterial growth, which was also dependent on the bacterial individuals. The growth promotion of most tested strains decreased after the corncob biochar was washed, which is probably due to the loss of nutrients in the fresh biochar [5]. Correspondingly, when amended with water-soluble compounds washed from biochar, the stimulation in cell growth could be observed for most strains. It is interesting that the positive impact on cell growth of *Pseudomonas* sp. and *B. cereus* by the unwashed biochar could be changed to a negative influence when water-soluble compounds were removed, presumably because the ash content in the fresh biochar was closely related with the bacterial growth [5]. However, as for *A. chroococcum*, *B. megaterium* ACCC01667, and *K. viridis*, greater growth promotions were obtained when the washed biochar was applied compared with the application of unwashed biochar, which might be attributed to the inhibition of water-soluble compounds in cell growth, as shown in Figure 4. All these results suggested that the cell growth in the presence of biochar was impacted not only by the large surface area and porous structure of biochar but also by the water-soluble compounds adsorbed on it.

### 3.4. Impact of Biochar on Potassium-/Phosphate-Solubilizing Bacteria

The effect of biochar on bacterial potassium-/phosphate-solubilizing activity was examined because the crop production was related with the increasing soil fertility induced by microbial activity [21]. Three strains, including *B. mucilaginosus*, *B. megaterium* (AS1217), and *Ochrobactrum* sp., with dramatic growth promotions induced by biochar, were selected to study their potassium-/phosphate-solubilizing activity after biochar addition. As shown in Table 2, the potassium-/phosphate-releasing contents varied significantly between the cultures with and without corncob biochar for all three tested bacteria. The released potassium content in the culture of *B. mucilaginosus* with biochar was 5.9 times higher than that without biochar. When compared with the control, the released phosphate content increased

by 2.5 times in the biochar culture of *B. megaterium*, but only increased by 0.24 times in the biochar culture of *Ochrobactrum* sp. The result is consistent with the growth variations observed in biochar added cultures, in which the cell amount was increased by 105.3%, 62.87% and 20.71% respectively for *B. mucilaginosus*, *B. megaterium* (AS1217), and *Ochrobactrum* sp. Therefore, biochar application could enhance the microbial activity through stimulating the cell growth.

**Table 2.** Impact of corncob biochar on potassium-/phosphate-releasing content of bacterial cultures.

| Characteristic | *Bacillus Mucilaginosus* AS1153 | | *Bacillus Megaterium* AS1217 | | *Ochrobactrum* sp. ACCC10085 | |
|---|---|---|---|---|---|---|
| | −biochar | +biochar | −biochar | +biochar | −biochar | +biochar |
| Potassium content ($\mu g \cdot mL^{-1}$) | 9.51 ± 0.43 | 67.27 ± 0.44 | — | — | — | — |
| Phosphate content ($\mu g \cdot mL^{-1}$) | — | — | 28.37 ± 0.65 | 98.98 ± 1.13 | 86.32 ± 0.89 | 107.11 ± 0.71 |

"−"/"+": Bacterial strains were cultured without or with biochar addition. "—": Not determined.

## 4. Discussion

Biochar is known to increase crop production through enhancing soil microbial activities. Anderson et al. [12] documented a biochar-induced result at the family level via the terminal restriction fragment length polymorphism (TRFLP) method, in which the pine-derived biochar showed a positive effect on the growth of Bradyrhizobiaceae, Hyphomicrobiaceae, Streptosporangineae, and Thermomonosporaceae, but a negative effect on Streptomycetaceae and Micromonosporaceae. However, little information is currently available on how biochar influences the growth and activities of soil biota at the species level [1]. Here, the effect of biochar addition on the growth of 12 soil bacterial strains belonging to different genus or species was investigated. To exclude possible interferences that soil may produce, the artificial medium added with biochar samples was applied. As a result, different species belonging to the same genus or not showed disparate responses to the same biochar sample. Moreover, such an alterable influence on bacterial growth was observed even in different strains from the same species (*B. megaterium* and *B. pumilus* in Figure 1), suggesting that the influence of biochar was strain-dependent (Figure 1). All three biochar samples greatly supported the growth of *B. mucilaginosus* AS1153, whereas the growth of *Kitasatospora viridis* was significantly blocked by biochar, indicating the potential activation/toxicity of organic compounds absorbed to biochar towards bacteria [22].

Previous studies have demonstrated that biochar generated from different feedstocks lead to different disturbances in soil biota community structure [15]. Similarly, Figure 1 revealed that biochar samples derived from rice husk, corncob, and bamboo exhibited different impacts on the growth of the same bacterial strain. Previous researches have suggested that changes in microbial community were mainly due to the large surface area and porous structure of biochar [19]. Both the surface area and average pore size varied considerably among rice husk biochar, corncob biochar, and bamboo biochar (Table 1), which might contribute to different responses of the same strain. Compared to rice husk biochar and bamboo biochar, corncob biochar had an excellent performance in promoting the growth of most bacterial strains, which might be due to the favorable physico-chemical properties of corncob biochar, such as high contents of TN, TOC, and volatile matter, and moderate pH value and surface area.

In general, biochar supplies a favorable habitat for bacteria growth through cell sorption [6]. However, the direct relationship between cell adsorption rate and growth stimulation has not been investigated, although biochar could adsorb all the tested strains at different extents (Figure 4). The maximal adsorption was observed on *B. thuringiensis* but the greatest cell growth promotion was detected on *B. mucilaginosus*. Interestingly, the growth of *B. mucilaginosus* could also be significantly improved after the inoculum was absorbed by the washed biochar (Figure 3). The similar phenomenon

also spread differentially across the other tested strains (data not shown), suggesting that biochar could be applied as a potential inoculum carrier. The adsorbing capability of biochar towards *K. viridis* was 25.1% but its growth was negatively influenced by biochar addition. It was presumed that besides the intrinsic structure of biochar, other factors might influence bacterial growth.

Water-soluble organic compounds (WSOCs), such as karrikins, carboxylic acids, phenolic and organic N compounds, and polycyclic aromatic hydrocarbons, have been proved to be available in biochar and could positively or negatively influence plant growth by affecting microbial-mediated processes in soil, such as C and N cycling [23–26]. Therefore, the influence of water-soluble components in biochar on cell growth was investigated. After soluble components were removed from biochar, the growth-promotion degree of most tested strains decreased except for *A. chroococcum* and *B. megaterium* (ACCC01667). Whereas the negative influence of washed biochar on cell growth was detected in *Pseudomonas* sp. and *B. cereus*, which is in accordance with the previous observation that the intrinsic properties of biochar could adversely impact bacterial growth [6]. Especially, the growth inhibition of *K. viridis* disappeared when the corncob biochar was washed (Figure 4). Correspondingly, the growth of *A. chroococcum*, *B. megaterium* (ACCC01667), and *K. viridis* was blocked by the water extracts of biochar (Figure 4). More recently, compounds inhibiting microbial activities have also been found either in biochar [7] or its water extracts after introduction into soil [27]. Besides, the water-soluble components in biochar promoted the growth of strain AS1217 but inhibited the other strain ACCC01667 belonging to *B. megaterium* (Figure 4). Therefore, it was unlikely that the impact of WSOCs spread equally across different phylotypes [9]. All these results indicated that water-soluble components in biochar could either stimulate or inhibit bacterial growth depending on individual bacteria, which was in line with the previous findings that some chemicals that commonly adsorbed in the biochar had the possible species-specific effect on a variety of microorganisms [28,29].

To further understand how biochar improves crop production by enhancing soil fertility, the influence of corncob biochar on potassium-/phosphate-solubilizing activities was also examined in this paper [21]. The K-solubilizing strain *B. mucilaginosus* could convert unavailable potassium to its available form through biological processes, which explained its plant growth promoting activity [12]. The present study showed that biochar could dramatically enhance the potassium-solubilizing activities of *B. mucilaginosus* culture through greatly increasing the viable cell count (Figure 1 and Table 2). To date few studies have dealt explicitly with biochar effects on P availability, but biochar could aid P mobilization by promoting the growth of bacteria involved with this [12]. In accordance with the previous findings, enhanced growth of two phosphate-solubilizing strains, *B. megaterium* (AS1217) and *Ochrobactrum* sp., lead to significant increases in phosphate-releasing content of bacterial cultures. Both the increased potassium and phosphate-releasing activities could obviously cause the increment of soil fertility [21].

## 5. Conclusions

The interaction of biochar and bacteria is subjected to many factors. Firstly, biochars derived from different feedstocks exhibit different impacts on the bacterial growth; due to favorable physico-chemical properties, corncob biochar had an excellent performance in promoting the growth of most of the tested bacterial strains. Secondly, the positive or negative effect of biochar on bacteria is proved to be taxon-dependent. Furthermore, besides the adsorption capability of biochar toward bacteria, which makes biochar a potential inoculum carrier, the water-soluble compounds in biochars could dramatically affect the bacterial growth, and such effect is also strain-specific.

**Author Contributions:** Conceptualization, F.Y., Y.Z. and X.L.; methodology, F.Y., W.L. and Y.Z.; software, W.L. and W.T.; validation, W.T., X.L. and W.C.; formal analysis, J.M.; investigation, F.Y., W.L. and Y.Z.; resources, J.M. and W.C.; data curation, F.Y.; writing—original draft preparation, F.Y. and Y.Z.; writing—review and editing, X.L.; supervision, X.L. and W.C.; project administration, X.L. and W.C.; funding acquisition, X.L. and W.C.

**Funding:** This research was supported by the National Key R&D Program of China (2017YFD0200800) and the National Natural Science Foundation of China (31401681).

**Conflicts of Interest:** The authors declare no conflicts of interest.

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
