# Peer review of "Strain-Specific Effects of Biochar and Its Water-Soluble Compounds on Bacterial Growth"

_applsci, doi:10.3390/app9163209_

Round 1
Reviewer 1 Report
This manuscript considers the influence of biochar obtained from different sources and its water-soluble compounds on the growth of bacteria, with the presented results suggesting a strain-specific influence of biochar and respective water-soluble compounds on bacteria growth. My general feeling is that the manuscript is worth publishing in Applied Sciences after properly addressing the following comments/questions/suggestions:
- Consider rewriting the title of the manuscript
Abstract
- “Previous studies have revealed the disturbance of biochar on microbial community at above family level.” – Rewrite the sentence
- “bacteria individuals” (and subsequent) – Replace by “individual bacteria”
- “suggesting the effect of biochar” – Replace by “suggesting that the effect of biochar”
- “that influences the cell growth” – Replace by “that influences cell growth”
Introduction
- Required English revision
- “Biochar was reported to have a highly positive or negative effect on bacterial community abundance [10].” - ¿?
- “Almost all these studies so far focused on the changes in microbial community at above family level, there was little information about the effects of biochar on bacteria at genus or species levels.” – Rewrite
- “on the bacteria growth.” – Replace by “on bacteria growth.”
Materials and methods
- “Biochar was produced from corncob, rice husk and bamboo …” – What about the source of these materials?
- “The ash content of biochar was determined by thermogravimetric analysis method [15].” – Conditions? That is, type of gas atmosphere, gas flow feeding, range of temperatures, heating rate, etc.
- Give numbers to the expressions. For instance:
Variation (%) = 100 × (Cbio – Cck)/Cck (1)
- “Overnight cultures of the tested strains (10 ml) were mixed with the sterilized biochar (60 mg) respectively and incubated at 30 ºC and 150 rpm for 3 h.” – Why these conditions? Are these conditions the result of previous optimization?
Results
- Section 3.1 title is too long. Consider simplifying it. Similar to subsequent sections and sub-sections.
- Table 1 – Average pore size not detected in the case of Bamboo biochar – Why? Any explanation?
- Figure 1/Figure 3 – Text is too small and can’t be read
- “The similar promotion in growth activity was also observed differentially on the other tested bacteria (data not shown).” – As data is not shown, can you at least mention a reference or some references?
- “Thus, the adsorption of corncob biochar towards bacteria probably is not the sole factor affecting bacterial growth.” – Please extend
Discussion (required English revision, especially in terms of the used verbs – past simple, past tense, present?)
- “Here, the effect of biochar addition on the growth of 12 soil bacterial strains belong to different genus or species was investigated.” – Replace by “Here, the effect of biochar addition on the growth of 12 soil bacterial strains belonging to different genus or species was investigated.”
- “As a result, different species belong to the same genus or not …” – Replace by “As a result, different species belonging to the same genus or not …”
- “All the three biochar samples …” – Replace by “All three biochar samples …”
- “Compared to rice husk biochar and bamboo biochar, corncob biochar had an excellent performance in promoting the growth of most bacterial strains, which is probably due to the favourable physico-chemical properties of corncob biochar.” – Which “favourable physico-chemical properties”?
Conclusions
- “the water-soluble compounds in “biochars could dramatically affect the bacterial growth, and such effect is also species-specific.” – Are you using “species-specific” or “strain-specific”?
Author Response
Dear Professor:
Thanks very much for your constructive comments. We have carefully revised the manuscript and double checked everything carefully in accordance with these comments. We hope every effort from us may clear up all your confusion on this work, and satisfy you at the same time.
Please see the attachment for our description on revision according to the comments.

Reviewer 2 Report
As authors mentioned, the influence of biochar and its water-soluble compounds on the growth of bacterial individuals was examined in the artificial culture condition, and the results suggested that the biochar together with its water-soluble compounds had a strain-specific impact on the bacteria growth. The general idea and experimental design for the method sounds good, but some of the data needs to be further clarified. The manuscript could be published in applied science after addressing the following issues: Authors mentioned that combination of biochar and bacteria successfully was demonstrated with a specific impact on the baceria growth just showing optical density. Please address properties of biochar used like BET surface area and SEM of biochar with bacteria for cell growth.Author Response
Dear Professor:
Thanks very much for your constructive comments. We have carefully revised the manuscript and double checked everything carefully in accordance with these comments. We hope every effort from us may clear up all your confusion on this work, and satisfy you at the same time.
Please see the attachment for our description on revision according to the comments.

Round 2
Reviewer 1 Report
I thank the authors for properly addressing all my comments/questions/suggestions.
Reviewer 2 Report
The authors addressed all queries. This manuscript might be acceptable for the publication.